# Ethnic differences in guideline-indicated statin initiation for people with type 2 diabetes in UK primary care, 2006–2019: A cohort study

**Sophie V. Eastwood**[1]*, **Rohini Mathur**[2], **Naveed Sattar**[3], **Liam Smeeth**[2], **Krishnan Bhaskaran**[2], **Nishi Chaturvedi**[1]

**1** University College London, London, United Kingdom, **2** London School of Hygiene &Tropical Medicine, London, United Kingdom, **3** University of Glasgow, Glasgow, United Kingdom

* sophie.eastwood@ucl.ac.uk

## Abstract

### Background

Type 2 diabetes is 2–3 times more prevalent in people of South Asian and African/African Caribbean ethnicity than people of European ethnicity living in the UK. The former 2 groups also experience excess atherosclerotic cardiovascular disease (ASCVD) complications of diabetes. We aimed to study ethnic differences in statin initiation, a cornerstone of ASCVD primary prevention, for people with type 2 diabetes.

### Methods and findings

Observational cohort study of UK primary care records, from 1 January 2006 to 30 June 2019. Data were studied from 27,511 (88%) people of European ethnicity, 2,386 (8%) people of South Asian ethnicity, and 1,142 (4%) people of African/African Caribbean ethnicity with incident type 2 diabetes, no previous ASCVD, and statin use indicated by guidelines. Statin initiation rates were contrasted by ethnicity, and the number of ASCVD events that could be prevented by equalising prescribing rates across ethnic groups was estimated. Median time to statin initiation was 79, 109, and 84 days for people of European, South Asian, and African/African Caribbean ethnicity, respectively. People of African/African Caribbean ethnicity were a third less likely to receive guideline-indicated statins than European people ($n/N$ [%]: 605/1,142 [53%] and 18,803/27,511 [68%], respectively; age- and gender-adjusted HR 0.67 [95% CI 0.60 to 0.76], $p < 0.001$). The HR attenuated marginally in a model adjusting for total cholesterol/high-density lipoprotein cholesterol ratio (0.77 [95% CI 0.69 to 0.85], $p < 0.001$), with no further diminution when deprivation, ASCVD risk factors, comorbidity, polypharmacy, and healthcare usage were accounted for (fully adjusted HR 0.76 [95% CI 0.68, 0.85], $p < 0.001$). People of South Asian ethnicity were 10% less likely to receive a statin than European people (1,489/2,386 [62%] and 18,803/27,511 [68%], respectively; fully adjusted HR 0.91 [95% CI 0.85 to 0.98], $p = 0.008$, adjusting for all covariates). We estimated that up to 12,600 ASCVD events could be prevented over the lifetimes of people currently affected by type 2 diabetes in the UK by equalising statin prescribing across ethnic groups. Limitations included incompleteness of recording of routinely collected data.

**Data Availability Statement:** Data were obtained from the CPRD (www.cprd.com). CPRD is a research service that provides primary care and

linked data for public health research. CPRD data governance and our own licence to use CPRD data do not allow us to distribute or make available patient data directly to other parties. Researchers can apply for data access with CPRD and must have their study protocol approved by the Independent Scientific Advisory Committee for Medicines and Healthcare products Regulatory Agency database research.

**Funding:** SVE is funded by a Diabetes UK Sir George Alberti Research Training Fellowship (grant number 17/0005588, https://www.diabetes.org.uk/ ). The funder had no role in the study design, data collection, analysis or interpretation of data, in drafting the manuscript or the decision to submit it for publication.

**Competing interests:** I have read the journal's policy and the authors of this manuscript have the following competing interests: RM reports a grant from the Wellcome Trust during the conduct of the study and personal fees from Amgen, outside the submitted work. NS has received grants and personal fees from Boehringer Ingelheim, and personal fees from Amgen, AstraZeneca, Eli Lilly, Merck Sharp & Dohme, Novartis, Novo Nordisk, Pfizer, and Sanofi outside the submitted work. LS has received grants from the British Heart Foundation and Diabetes UK during the conduct of the study, grants from the Wellcome Trust, Medical Research Council, National Institute for Health Research, GlaxoSmithKline, British Heart Foundation, Diabetes UK, the Newton Fund and UKRI, outside the submitted work, and is a non-executive director of the MHRA. KB has received grants from the Wellcome Trust and the Royal Society during the conduct of this study. NC has received grants from Diabetes UK during the conduct of this study and personal fees from AstraZeneca outside of the submitted work. SVE has no competing interests to declare.

**Abbreviations:** ASCVD, atherosclerotic cardiovascular disease; COPD, chronic obstructive pulmonary disease; CPRD, Clinical Practice Research Datalink; HDL, high-density lipoprotein; IMD, Index of Multiple Deprivation; NICE, National Institute for Health and Care Excellence; QOF, Quality and Outcomes Framework; TC/HDL, total cholesterol/high-density lipoprotein cholesterol ratio; VIF, variance inflation factor.

## Conclusions

In this study we observed that people of African/African Caribbean ethnicity with type 2 diabetes were substantially less likely, and people of South Asian ethnicity marginally less likely, to receive guideline-indicated statins than people of European ethnicity, even after accounting for sociodemographics, healthcare usage, ASCVD risk factors, and comorbidity. Underuse of statins in people of African/African Caribbean or South Asian ethnicity with type 2 diabetes is a missed opportunity to prevent cardiovascular events.

## Author summary

### Why was this study done?

- People of South Asian and African/African Caribbean ethnicity living in the UK are more likely to have type 2 diabetes than people of European ethnicity, and have higher rates of cardiovascular complications, e.g., heart attacks and strokes.

- Lowering blood cholesterol with statin treatment reduces cardiovascular complications, but previous studies suggest ethnic differences exist in statin prescribing for people with diabetes.

- However, no study has sought explanations for identified ethnic differences, or accounted for changes in prescribing guidelines, so we aimed to provide a timely, representative assessment of ethnic differences in guideline-indicated statin prescribing in type 2 diabetes.

### What did the researchers do and find?

- We identified people with newly diagnosed type 2 diabetes in primary care eligible for statin treatment for primary prevention of cardiovascular disease, then compared rates of statin initiation for people of European, South Asian, and African/African Caribbean ethnicity.

- People of African/African Caribbean ethnicity with type 2 diabetes were 24% less likely to receive guideline-indicated statin treatment than people of European ethnicity, and people of South Asian ethnicity 9% less likely.

- Ethnic differences remained after allowing for differences in cholesterol levels, other cardiovascular risk factors, demographic factors, deprivation, healthcare usage, comorbidity, and polypharmacy.

- We estimated that equalising statin prescribing rates across these 3 ethnic groups would prevent up to 12,600 heart attacks and strokes over the lifetimes of people currently affected by type 2 diabetes in the UK.

## What do these findings mean?

- People with type 2 diabetes of African/African Caribbean and South Asian ethnicities are less likely to be prescribed a statin for primary prevention of cardiovascular disease in the UK than people of European ethnicity.

- Policies to increase statin use among people of African/African Caribbean and South Asian ethnicity with type 2 diabetes could substantially reduce the excess burden of cardiovascular events in these groups.

## Introduction

Type 2 diabetes is a potent risk factor for cardiovascular disease, itself causing 17.9 million deaths worldwide per year, 80% of which are due to myocardial infarction and stroke combined [1]. People of South Asian and African/African Caribbean ethnicity living in the UK experience not only a 3-fold higher prevalence of type 2 diabetes [2], but higher rates of the atherosclerotic cardiovascular disease (ASCVD, i.e., coronary heart disease and stroke) complications of diabetes than their European-origin counterparts [3]. Explanations for this are unclear but may relate to poorer control of modifiable ASCVD risk factors, such as hypercholesterolaemia, in South Asian and African/African Caribbean groups [4–6]. Statin treatment is a crucial tool in reducing ASCVD risk in diabetes; the Collaborative Atorvastatin Diabetes Study (CARDS) trial demonstrated a 37% reduction in ASCVD events over 4 years of follow-up [7]. In the UK, statin prescribing in people with diabetes occurs in primary care and is governed by the National Institute for Health and Care Excellence (NICE) guidelines [8–10]. NICE guidelines have varied by time period regarding thresholds for statin prescription; the 2006 guideline designated type 2 diabetes a secondary prevention state and thus advocated universal prescription [9], the 2008 guideline recommended statin use in all ≥40 years old or with a 10-year ASCVD risk of ≥20% or with ≥1 other ASCVD risk factor [8], whilst the 2014 guideline advised high-intensity statin prescription (e.g., atorvastatin 20 mg) simply for all with a 10-year ASCVD risk of ≥10% [10] (diabetes at this point being a multiplier in the recommended ASCVD risk equation).

Despite ethnic differences in diabetes risk and resultant ASCVD risk, as far as we know only 2 UK studies have examined ethnic differences in statin prescribing for primary prevention in diabetes. In a 2007 study of 36 practices in South London, Millett et al. [5] described lower rates of statin prescription in people with diabetes of black African origin than in their white counterparts (49% versus 64%), though prescribing was generally similar for South Asian and white people. Comparable findings were reported from a 2007 study based on data from the Health Survey for England [6]. Furthermore, several US studies of health record data indicate substantial underprescribing of statins to African American people with diabetes in whom statins are indicated, in comparison to their white counterparts [11–14], though we are unaware of any similar North American data for people of South Asian ethnicity. Evidence suggests statins are equally effective in people of European, South Asian, and African/African Caribbean ethnicity [15], so there is no rationale for restricting their use on the basis of ethnicity.

However, to our knowledge, there are no studies of ethnic differences in statin prescribing for type 2 diabetes using nationally representative datasets, exploring reasons for prescribing differences, restricting analysis to people with guideline indications for statin use (as opposed

to everyone with diabetes), or employing data from after the 2014 NICE guideline changes in statin prescribing [10].

Therefore, using a nationally representative database of prescribing records, we aimed to study ethnic differences in statin initiation for the statin-eligible (according to guidelines [10]) population with type 2 diabetes. Specifically, we (i) contrasted statin initiation rates for people of European, South Asian, and African/African Caribbean ethnicity, (ii) sought explanations for differences found, and (iii) estimated the population impact of equalising statin prescribing for the UK's 3 main ethnic groups.

## Methods

### Study population

We used data from the Clinical Practice Research Datalink (CPRD), a national database of over 12 million anonymised primary care records from 836 practices [16]. CPRD is representative of the UK population with regards to age, gender, and ethnicity [16]. Derivation of the study cohort required several steps (Table 1). First, we applied a diabetes diagnostic adjudication algorithm, based on a previously published version [17], to the population of individuals in CPRD with European, South Asian, or African/African Caribbean ethnicity Read codes (see below) to obtain those with type 2 diabetes. We then limited this to an adult (≥18 years) cohort of individuals whose first diabetes diagnostic or medication code (i.e., their index date) was after 1 January 2006 and over 12 months after the latest of (i) the patient's current registration date or (ii) the practice's up to CPRD standard date. This was to ensure an 'incident' cohort, i.e., to avoid including prevalent cases of diabetes that were recorded soon after the individual joined a new practice or the practice records met CPRD standards. We then excluded people with prevalent ASCVD or statin use at the time of diabetes diagnosis. Following this, we applied time-period-dependent criteria to select the population in whom statins were indicated, using the 2006 NICE guideline [9] for people with index dates between 1 January 2006 and 31 May 2008 (all of whom were eligible for statin prescription, and thus inclusion), the 2008 guideline [8] for people with index dates between 1 June 2008 and 31 July 2014 (who were eligible for statin prescription, and thus inclusion, if they were ≥40 years old or had a 10-year ASCVD risk of ≥20% or had ≥1 other ASCVD risk factor), and the 2014 guideline [10] for people with index dates between 1 August 2014 and 30 June 2019 (who were eligible for statin prescription, and thus inclusion, if they had a 10-year ASCVD risk of ≥10%) (see Table 1). For the third time period, where the criterion for statin indication changed to a 10-year ASCVD risk score ≥ 10%, the index date was changed to the date of the ASCVD risk score ≥ 10% record. People with prevalent ASCVD or statin use at their new index date were once again excluded at this stage. The intention of examining statin initiation by time-varying eligibility was to enhance the validity of the study by selecting a denominator sample for each period that reflected appropriate prescribing at the time.

### Outcome

Statin initiation was designated by a first ever statin prescription date on or after the index date (i.e., the date when follow-up started). In all cases (i.e., for each of the 3 time periods pertaining to 2006, 2008, and 2014 NICE guidelines), the index date was the date when the person became eligible for statin treatment, according to the relevant guideline in use at the time (see above). By using the first date of eligibility for statin treatment as the index date (irrespective of what factors governed that eligibility), we aimed to produce a focused outcome of appropriate prescribing. This enabled a fair comparison for the 3 ethnic groups across all time periods of the proportion of those who should have received a statin who actually did so.

**Table 1. Derivation of cohort, by ethnicity.**

| European/South Asian/African/African Caribbean ethnicity codes in CPRD, *n* = 5,566,120 | | |
|---|---|---|
| 4,968,447 European (89%), 363,735 South Asian (7%), 233,938 African/African Caribbean (4%) | | |
| ↓ | | |
| **Algorithm-adjudicated T2DM after 1 January 2006, *n* = 158,554** | | |
| 138,370 European (87%), 13,850 South Asian (9%), 6,334 African/African Caribbean (4%) | | |
| ↓ | | |
| **Adult (≥18 years), incident cohort, *n* = 102,436** | | |
| 93,120 European (91%), 6,499 South Asian (6%), 2,817 African/African Caribbean (3%) | | |
| ↓ | | |
| **Exclude people with prevalent statin use or ASCVD, *n* = 56,291** | | |
| 49,724 European (88%), 4,451 South Asian (8%), 2,116 African/African Caribbean (4%) | | |
| ↓ | | |
| **T2DM index date 1 Jan 2006 to 31 May 2008, *n* = 10,773** 9,934 European (92%), 582 South Asian (5%), 257 African/African Caribbean (2%) ↓ | **T2DM index date 1 Jun 2008 to 31 Jul 2014, *n* = 30,512** 26,799 European (88%), 2,544 South Asian (8%), 1,169 African/African Caribbean (4%) ↓ | **T2DM index date 1 Aug 2014 to 30 Jun 2019, *n* = 15,006** 12,991 European (87%), 1,325 South Asian (9%), 690 African/African Caribbean (5%) ↓ |
| **Statins indicated (NICE 2006 [9]) in all, for 'secondary prevention', *n* = 10,773** 9,934 European (92%), 582 South Asian (5%), 257 African/African Caribbean (2%) ↓ | **Statins indicated (NICE 2008 [8]) in all ≥40 years or 10-year ASCVD risk ≥ 20% or 1+ risk factors (family history of ASCVD, 'at risk' ethnicity, dyslipidaemia, hypertension, obesity, microvascular disease), *n* = 30,235** 26,522 European (88%), 2,544 South Asian (8%), 1,169 African/African Caribbean (4%) ↓ | **Statins indicated (NICE 2014 [10]) in all with 10-year ASCVD risk ≥ 10%, *n* = 5,352** 4,882 European (91%), 328 South Asian (6%), 142 African/African Caribbean (3%) ↓ |
| | | **ASCVD risk score date 0–5 years preceding T2DM index date, *n* = 2,974** 2,716 European (91%), 175 South Asian (6%), 83 African/African Caribbean (3%) ↓ | **ASCVD risk score date subsequent to T2DM index date, *n* = 2,378\*** 2,169 European (91%), 151 South Asian (6%), 58 African/African Caribbean (2%) ↓ **Exclude people with prevalent statin use or ASCVD at ASCVD risk score date, *n* = 2,000** 1,833 European (92%), 119 South Asian (6%), 48 African/African Caribbean (2%) ↓ |
| ***n* = 45,925** | | |
| 40,963 European (89%), 3,413 South Asian (7%), 1,549 African/African Caribbean (3%) | | |
| ↓ | | |
| **Complete case analysis, *n* = 31,039** | | |
| 27,511 (89%) European, 2,386 South Asian (8%), 1,142 African/African Caribbean (4%) | | |

ASCVD, atherosclerotic cardiovascular disease; CPRD, Clinical Practice Research Datalink; NICE, National Institute for Health and Care Excellence; T2DM, type 2 diabetes mellitus.

\*These people were assigned a new index date equivalent to their ASCVD risk score date, and those with prevalent ASCVD/statin use at the new index date were excluded.

Other lipid-lowering medications were not considered, since they are generally second-line drugs (e.g., ezetimibe, for which there were very small numbers of incident prescriptions in people not on statins: *n* = 38, *n* < 5, and *n* < 5 for people of European, South Asian, and African/African Caribbean ethnicity, respectively) or their use falls outside of all NICE guidelines

pertinent to the study, and thus they are rarely initiated in primary care. We did not examine ethnic differences in statin prescribing for secondary prevention. This is an important, but different question and would have required a different approach, as initiation of statins for secondary prevention most commonly occurs in secondary care, making it difficult to establish index dates of statin eligibility and initial prescription.

## Covariates

The main exposure in this study was self-reported (i.e., assigned by the patient themselves) European, South Asian, or African/African Caribbean ethnicity. Ethnicity is a complex construct that groups people together who identify with each other through a shared culture, and encompasses factors in common such as cultural heritage, ancestry, language, history, diet, or religion [18,19]. Thus, it should be self-defined, according to the group the individual most identifies with [20]. Ethnicity was designated using ethnicity Read codes, usually recorded by primary care staff during a patient's initial registration with a practice. Primary care practices were financially incentivised to record ethnicity, under the Quality and Outcomes Framework (QOF) [21], from 2006 to 2011. A previously published algorithm [22] adjudicated on cases of multiple or differing codes per patient. This algorithm was also used to derive the following ethnic sub-groups for sub-group analyses: British, Irish, other white, Indian, Pakistani, Bangladeshi, other South Asian, Caribbean, African, and other black (S1 Text). People with Read codes indicating mixed ethnicity were excluded from this study.

Covariates were selected pragmatically (i.e., governed by what was available in the dataset) to enable investigation of areas of potential confounding—sociodemographics (age, gender, socioeconomic deprivation), ASCVD risk factors (smoking, total cholesterol/high-density lipoprotein [HDL] cholesterol ratio [TC/HDL], BMI, antihypertensive use), healthcare usage (consultation rate), comorbidity (4 chronic disease areas), and polypharmacy (number of medications)—and to enable investigation of effect modification by degree of hyperlipidaemia, geographical location (London versus non-London practice and country within the UK—examined in separate sets of models), guideline time period, ethnic sub-group, and gender. Age at index date was calculated by subtracting year of birth from year of index date. Gender was self-reported male or female (no other categories were present in the dataset). Practice location was studied by (i) London versus non-London practices and (ii) country within the UK. Postcode-designated, quintilised Index of Multiple Deprivation (IMD) [23] data were available at the practice level for the whole cohort, and at the individual level for 58%. Smoking was categorised as never, ex, or current, based on Read codes up to 5 years prior to and preferentially closest to the index date. Healthcare usage was categorised by quartile of the number of patient-initiated consultations in the year before the index date, and polypharmacy by quartile of the number of different medications prescribed in the year before the index date. Values of $HbA_{1c}$, BMI, and total and HDL cholesterol closest to the index date, in the 5 years preceding it, were used. TC/HDL and non-HDL (total cholesterol minus HDL cholesterol) values were derived from the recorded total and HDL cholesterol values. Prevalent comorbidity (cancer, asthma/chronic obstructive pulmonary disease [COPD], chronic kidney disease [CKD], and serious mental illness) and antihypertensive use were determined at index date. For those who did not receive a statin throughout follow-up, presence of a recorded reason for statin declinature was noted (this variable consisted of the presence of 1 or more of the following codes in the health record: statin contraindicated [3%], statin declined [80%], statin not tolerated [21%], statin purchased over the counter [1%]). Finally, Read codes indicating exception reporting (i.e., exclusion) from the diabetes QOF (the system by which primary care physicians are incentivised to record treatment metrics on their patients) were noted.

## Statistical analysis

All analytic methods were pre-specified in a scientific protocol (S1 Protocol).

Complete case analysis (i.e., inclusion of data only from people with complete recording of all baseline covariates) was used for all baseline covariates in the main models. We chose this approach over multiple imputation as the missing at random assumption (necessary for multiple imputation) may not hold in primary care records (e.g., BMI is more likely to be recorded if not in the healthy range), and complete case analysis has been proven to be valid providing the missingness is conditionally independent of the outcome [24]. Of note, most covariates of interest were complete in our data: Ethnicity was required for inclusion in the study population; age, gender, and deprivation had no missing data; and other variables were defined by presence/absence of specific codes in the health record (e.g., specific diagnoses or medications, healthcare usage), so could not be missing. The only variables with missing data were therefore smoking status, TC/HDL, and BMI, which relied solely on physicians' recording, and thus were not complete for all individuals.

Baseline characteristics were compared by ethnicity, adjusting means for age, or using proportions age-standardised to the CPRD population. Additionally, characteristics, where available, were compared between the complete case analysis sample and the sample for which some data were missing, to explore important differences between those with and without missing data.

The pre-specified primary comparison of statin initiation in people of South Asian or African/African Caribbean versus European ethnicity was studied using multilevel Cox proportional hazards models with robust standard errors to account for within-practice clustering of prescribing patterns. Prescribing behaviour is similar for doctors working in the same practice [25] (i.e., clustering of prescribing patterns), resulting in individuals attending the same practice not being independent of each other with respect to statin initiation, which may in turn lead to underestimation of standard errors, i.e., excess type 1 errors. To counteract this, we fitted models using robust standard errors, with practice as the cluster variable. Follow-up was censored at the earliest date of the following: statin initiation, the study end date, death, or leaving a CPRD practice. Models were adjusted for age and gender, plus additional factors in turn separately (deprivation, smoking, healthcare usage, TC/HDL, BMI, comorbidity [comprising cancer, asthma/COPD, CKD, and serious mental illness, entered as separate individual terms], polypharmacy, and antihypertensive use), then finally all factors together. We evaluated the likelihood of the results indicating a true ethnic difference chiefly via inspection of the 95% confidence intervals of a given association [26], generated using standard errors from the 2-sided z test from Cox regression models.

The proportional hazards assumption was verified by inspection of log–log survival plots by ethnic group and performing a formal test of Schoenfeld residuals. Collinearity was investigated between variables for the model containing all covariates by calculating variance inflation factors (VIFs), and variables with VIF $\geq 10$ were excluded in sensitivity analyses to assess the impact on standard errors.

Pre-specified secondary sub-group analyses were performed, by inspecting both within-group HRs and ethnicity × sub-group interaction terms, to study whether ethnic differences in statin prescribing varied by (i) baseline TC/HDL tertile, (ii) London practice location, (iii) time period in relation to NICE guidelines (i.e., 1 January 2006–31 May 2008, 1 June 2008–31 July 2014, or 1 August 2014–30 June 2019), (iv) ethnic sub-group, (v) age group at index date ($\leq 45$ years, $>45$ to $\leq 65$ years, or $>65$ years), (vi) country (England, Scotland, and Wales only, due to small numbers in Northern Ireland), and (vii) gender. Where major differences in statin initiation were shown, we compared baseline characteristics between sub-groups. Pre-specified secondary sensitivity analyses explored the impact of adjustment for further factors—HbA$_{1c}$, patient-level IMD, non-HDL cholesterol, and BMI categories (using cut points of

25 kg/m$^2$ to designate overweight and 30 kg/m$^2$ to designate obesity in people of European ethnicity, and cut points of 23 kg/m$^2$ and 27.5 kg/m$^2$ in people of South Asian or African/African Caribbean ethnicity [27])—or exclusion of people who gave a reason for declining a statin or who were excepted from the diabetes QOF.

Finally, we estimated the population impact of equalising statin prescribing across ethnic groups by comparing the number of ASCVD events prevented at observed statin initiation rates for people of European ethnicity with those at observed statin initiation rates for people of South Asian and African/African Caribbean ethnicity, using estimated populations of people with type 2 diabetes in the latter 2 ethnic groups as the denominators, and a range of assumptions regarding statin potency and adherence.

This study is reported as per the Strengthening the Reporting of Observational Studies in Epidemiology (STROBE) guideline (S1 Checklist).

Analyses were carried out using Stata, version 16.

### Ethical approval

Ethical approval for this study was obtained from the Independent Scientific Advisory Committee (protocol 19_045, see protocol in S1 Protocol) and the London School of Hygiene & Tropical Medicine (project ID 14222).

### Patient and public involvement

There was no patient or public involvement in developing the research question, designing or conducting the study, interpreting the findings, or disseminating the results.

## Results

The study population comprised 27,511 (88%) people of European ethnicity, 2,386 (8%) people of South Asian ethnicity, and 1,142 (4%) people of African/African Caribbean ethnicity who were eligible for guideline-indicated statin initiation (Table 1), with a median follow-up time of 0.7 years (interquartile range 0.07–2.44 years). People with missing ethnicity ($n = 25,847$) were excluded from the study population, but comparisons with the European ethnic group are presented in S4 and S5 Tables. Statin initiation rates were lower in people of South Asian and African/African Caribbean than European ethnicity ($n$ [%] and rate per 1,000 person-years at risk: 1,489 [62%] and 0.36 [95% CI 0.34 to 0.37] for people of South Asian ethnicity, 605 [53%] and 0.24 [95% CI 0.22 to 0.26] for people of African/African Caribbean ethnicity, and 18,803 [68%] and 0.40 [95% CI 0.39 to 0.41] for people of European ethnicity). Median time to statin initiation was 79 (IQR 7–368) days for people of European ethnicity, and 109 (IQR 6–418) and 84 (IQR 2–520) days for South Asian and African/African Caribbean groups, respectively.

Mean age at diabetes onset was markedly lower in people of South Asian and African/African Caribbean ethnicity than those of European ethnicity (50 years and 52 years versus 59 years, respectively), and higher proportions of the former groups attended a practice in London and in the most deprived IMD quintile (Table 2). However, ASCVD risk factors were in general more favourable for the South Asian and African/African Caribbean groups than the European group (notably, lower levels of baseline cholesterol)—the exception to this was antihypertensive use, which was far higher in people of African/African Caribbean origin than in the other groups. There were no consistent ethnic differences in healthcare usage or comorbidity. After exclusion of people with missing data for TC/HDL ($n = 10,276$, 22%), BMI ($n = 3,723$, 9%), and smoking ($n = 887$, 3%), 68% of the sample ($n = 31,039$) remained for the complete case analysis (Table 1). A comparison of baseline characteristics by inclusion versus

**Table 2. Baseline characteristics of people with incident type 2 diabetes in 2006 or later with no prior statin use or ASCVD, by ethnicity (complete case analysis).**

| | European ethnicity | South Asian ethnicity | African/African Caribbean ethnicity |
|---|---|---|---|
| *N* (%) | 27,511 (88) | 2,386 (8) | 1,142 (4) |
| **Ethnic sub-group** | British: 26,238 (95); Irish: 257 (1); other/not stated white: 1,016 (4) | Indian: 932 (39); Pakistani: 628 (26); Bangladeshi: 213 (9); other/not stated South Asian: 612 (26) | Caribbean: 420 (37); African: 553 (48); other/not stated black: 170 (15) |
| **Age, years** | 59 ± 12 | 50 ± 11 | 52 ± 11 |
| **Age group** | | | |
| ≤45 years | 4,009 (15) | 888 (37) | 307 (27) |
| >45 to ≤65 years | 15,505 (56) | 1,276 (53) | 679 (59) |
| >65 years | 7,997 (29) | 222 (9) | 156 (14) |
| **Gender** | | | |
| Male | 15,249 (53) | 1,307 (53) | 589 (51) |
| Female | 12,262 (47) | 1,079 (47) | 553 (49) |
| **Practice location** | | | |
| Non-London | 24,861 (91) | 1,467 (66) | 440 (44) |
| London | 2,650 (9) | 919 (34) | 702 (56) |
| **Country** | | | |
| England | 19,205 (68) | 2,163 (90) | 1,093 (95) |
| Scotland | 5,464 (20) | 128(4) | 25 (2) |
| Wales | 2,540 (10) | 93(5) | 21 (3) |
| Northern Ireland | 302 (2) | 2 (0.01) | 3 (0.01) |
| **Deprivation—practice IMD quintile** | | | |
| 1 (least deprived) | 3,766 (12) | 203 (7) | 46 (3) |
| 2 | 4,713 (15) | 287 (11) | 108 (10) |
| 3 | 5,130 (18) | 505 (19) | 182 (15) |
| 4 | 6,118 (22) | 700 (26) | 360 (30) |
| 5 (most deprived) | 7,784 (32) | 691 (37) | 446 (42) |
| **Deprivation—patient IMD quintile*** | | | |
| 1 (least deprived) | 2,754 (15) | 184 (9) | 30 (3) |
| 2 | 3,230 (18) | 291 (16) | 79 (8) |
| 3 | 3,089 (20) | 334 (17) | 158 (17) |
| 4 | 3,141 (22) | 411 (29) | 233 (26) |
| 5 (most deprived) | 3,107 (24) | 464 (30) | 367 (46) |
| **Smoking** | | | |
| Never | 9,829 (38) | 1,510 (61) | 665 (58) |
| Ex | 12,721 (39) | 567 (23) | 350 (29) |
| Current | 4,961 (23) | 309 (16) | 127 (13) |
| **Number of consultations in previous year** | 7 (4–12) | 6 (3–10) | 6 (3–10) |
| **Number of consultations in previous year—quartiles** | | | |
| 1 (0 to 3 consultations) | 5,901 (21) | 607 (27) | 309 (24) |
| 2 (4 to 7 consultations) | 8,540 (29) | 772 (29) | 373 (28) |
| 3 (8 to 12 consultations) | 6,702 (24) | 577 (29) | 250 (19) |
| 4 (13+ consultations) | 6,368 (25) | 430 (16) | 210 (29) |
| **Total HDL cholesterol, mmol/l** | 4.96 ± 1.59 | 4.67 ± 1.45 | 4.21 ± 1.43 |
| **Non-HDL cholesterol, mmol/l** | 4.38 ± 1.18 | 4.16 ± 1.04 | 3.97 ± 1.12 |
| **HbA$_{1c}$, percent**** | 7.8 ± 2.1 | 7.6 ± 2.0 | 7.8 ± 2.3 |

*(Continued)*

**Table 2.** (Continued)

| | European ethnicity | South Asian ethnicity | African/African Caribbean ethnicity |
|---|---|---|---|
| **HbA$_{1c}$, mmol/mol**$^{**}$ | 61 ± 23 | 59 ± 23 | 60 ± 25 |
| **BMI, kg/m$^2$** | 34 ± 7 | 28 ± 6 | 31 ± 6 |
| Underweight/normal | 2,262 (9) | 183 (12) | 34 (5) |
| Overweight$^{***}$ | 7,074 (21) | 679 (25) | 205 (17) |
| Obese$^{***}$ | 18,175 (71) | 1,524 (63) | 903 (78) |
| **CKD** | 1,058 (4) | 41 (3) | 50 (6) |
| **Cancer** | 1,902 (6) | 48 (2) | 56 (6) |
| **Asthma/COPD** | 4,175 (17) | 307 (17) | 131 (12) |
| **Serious mental illness** | 1,405 (7) | 74 (4) | 63 (7) |
| **Number of different medications prescribed in previous year** | 6 (3–10) | 5 (2–9) | 5 (2–9) |
| **Number of different medications prescribed in previous year—quartiles** | | | |
| 1 (0 to 2 medications) | 5,527 (20) | 605 (22) | 287 (23) |
| 2 (3 to 5 medications) | 7,607 (27) | 620 (22) | 333 (25) |
| 3 (6 to 9 medications) | 6,929 (25) | 552 (29) | 267 (21) |
| 4 (10+ medications) | 7,448 (28) | 609 (28) | 255 (31) |
| **Antihypertensive use** | 12,477 (36) | 620 (29) | 516 (41) |
| **Recorded reason for statin declinature (if statin not prescribed)**$^{****}$ | 1,181/8,708 (10) | 66/897 (6) | 30/537 (4) |
| **Exception reported from diabetes Quality and Outcomes Framework** | 1,186 (6) | 102 (7) | 54 (5) |
| **Time period of index date, by NICE guideline** | | | |
| 1 Jan 2006 to 31 May 2008 | 5,583 (20) | 339 (14) | 167 (15) |
| 1 Jun 2008 to 31 Jul 2014 | 17,849 (65) | 1,791 (75) | 856 (75) |
| 1 Aug 2014 to 30 Jun 2019 | 4,079 (15) | 256 (11) | 119 (10) |

Data are n (age-standardised percent), n/N (age-standardised percent), age-adjusted mean ± SD, or median (IQR).

ASCVD, atherosclerotic cardiovascular disease; BMI, body mass index; CKD, chronic kidney disease; COPD, chronic obstructive pulmonary disease; HbA$_{1c}$, glycosylated haemoglobin A$_{1c}$; HDL, high-density lipoprotein; IMD, Index of Multiple Deprivation; NICE, National Institute for Health and Care Excellence.

$^*$Data available for a subset of 15,321 people of European ethnicity, 1,684 people of South Asian ethnicity, and 867 people of African/African Caribbean ethnicity.

$^{**}$Data available for a subset of 16,889 people of European ethnicity, 1,669 people of South Asian ethnicity, and 769 people of African/African Caribbean ethnicity.

$^{***}$BMI cut points for overweight and obesity, respectively, were 25 kg/m$^2$ and 30 kg/m$^2$ for people of European ethnicity and 23 kg/m$^2$ and 27.5 kg/m$^2$ for people of South Asian or African/African Caribbean ethnicity.

$^{****}$Reasons included the following: statin contraindicated, statin not tolerated, statin declined, or statin purchased over the counter.

exclusion in the complete case analysis demonstrated no obvious incongruities, except for lower consulting rates and antihypertensive usage in the latter group (S1 Table). Importantly, missingness did not differ greatly by ethnicity (33% for the European, 31% for the South Asian, and 28% for the African/African Caribbean group).

In age- and gender-adjusted models of ethnic differences in statin initiation for the guideline-eligible population, people of African/African Caribbean ethnicity were 33% less likely than people of European ethnicity to receive statin treatment (HR 0.67 [95% CI 0.60 to 0.76], $p < 0.001$), and people of South Asian ethnicity were 12% less likely (0.88 [95% CI 0.82 to 0.93], $p < 0.001$) (Fig 1). These differences were unaffected by further adjustment for deprivation, smoking, healthcare usage, BMI, comorbidity, or polypharmacy. However, accounting for baseline TC/HDL reduced ethnic differences (South Asian versus European, HR 0.93 [95%

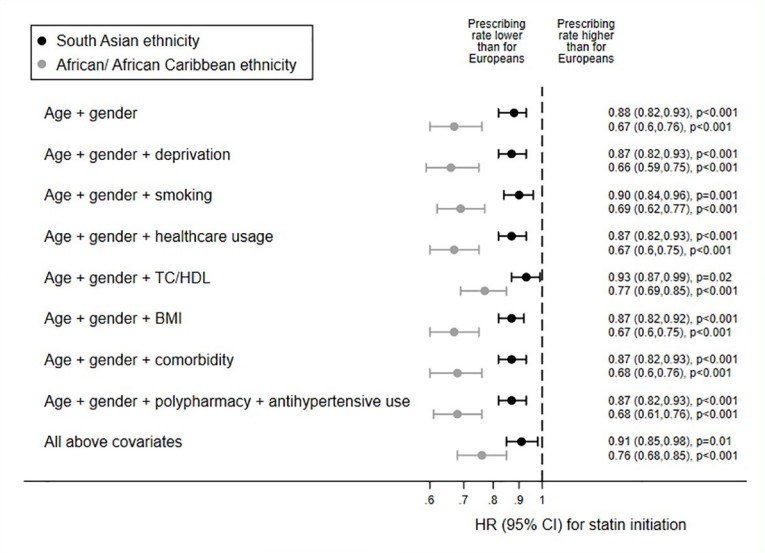

**Fig 1. Associations between ethnicity and guideline-indicated statin initiation after type 2 diabetes diagnosis.**
Data are HRs (marker) and 95% CIs (capped lines), from multilevel models accounting for intra-practice clustering, European ethnicity = referent category, i.e., HR for people of European ethnicity = 1. TC/HDL, total cholesterol/high-density lipoprotein cholesterol ratio.

CI 0.87 to 0.99], $p = 0.02$; African/African Caribbean versus European, HR 0.77 [95% CI 0.69 to 0.85], $p < 0.001$), with no further attenuation seen when all available baseline covariates were added to the model (HR 0.91 [95% CI 0.85 to 0.98], $p = 0.01$, and HR 0.76 [95% CI 0.68, 0.85], $p < 0.001$, respectively).

Secondary sub-group analyses showed that ethnic differences in statin initiation appeared to be more pronounced at the lowest and, for people of African/African Caribbean ethnicity, highest tertile of TC/HDL (African/African Caribbean ethnicity × TC/HDL tertiles 2 and 3, interaction $p = 0.15$ and $p = 0.63$, respectively; Fig 2). People of African/African Caribbean ethnicity in the highest TC/HDL tertile were more likely to be younger, with more adverse ASCVD risk factors (S2 Table). Ethnic differences in statin initiation seemed to be greatest for people attending practices outside London—indeed, for South Asian people attending practices in London, prescribing rates equalled those of European people (location × ethnicity, interaction $p = 0.04$; Fig 2). Similarly, ethnic differences appeared less marked for the last 5 years of the study, though there was no statistical evidence of interaction by calendar period (ethnicity × last time period, interaction $p = 0.24$ for South Asian people and $p = 0.51$ for African/African Caribbean people); these findings should be interpreted with caution as confidence intervals were very wide due to low power, and do in fact include the larger differences present for the prior periods. Of note, risks of underprescription appeared more distinct in certain ethnic sub-groups, i.e., for the 'other white' compared to the British group, for people of Bangladeshi compared to Indian or Pakistani ethnicity, and for the 'other black' compared to the Caribbean or African groups. Comparison of baseline characteristics between the sub-groups demonstrated that groups with lower statin initiation rates were, in general, more likely to be younger and more deprived, with higher TC/HDL and less healthcare utilisation (S3 Table).

Results of further secondary sub-group analyses by age group, country, and gender (S1 Fig), and of sensitivity analyses adjusting for HbA$_{1c}$, patient-level deprivation, or BMI category or excluding those with recorded reasons for statin declinature or who had been exception

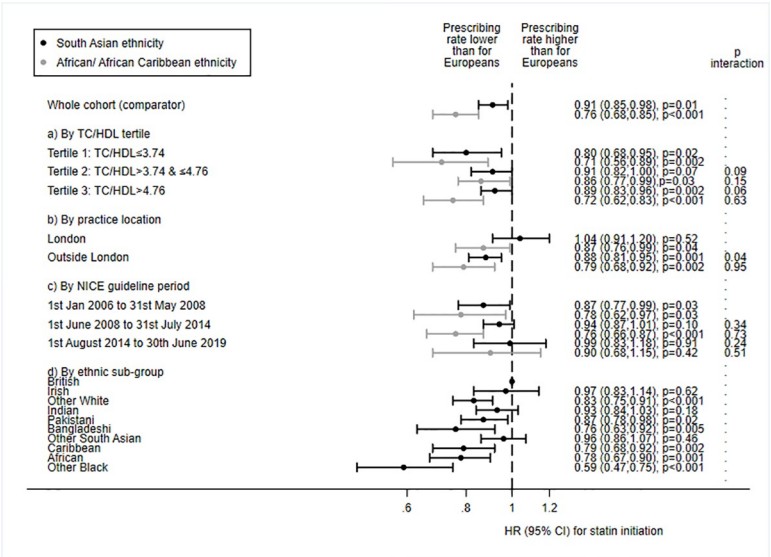

**Fig 2. Associations between ethnicity and guideline-indicated statin initiation after type 2 diabetes diagnosis: Sub-group analyses.** Sub-group analyses by (a) total cholesterol/high-density lipoprotein cholesterol ratio (TC/HDL) tertile, (b) London practice location, (c) time period of National Institute for Health and Care Excellence (NICE) guidelines, and (d) ethnic sub-group. Data are HRs (marker) and 95% CIs (capped lines) adjusted for age, gender, deprivation, smoking, healthcare usage, TC/HDL, BMI, prevalent comorbidity, medication usage, and antihypertensive usage, from multilevel models accounting for intra-practice clustering. European ethnicity = referent category, i.e., HR for people of European ethnicity = 1.

reported from the diabetes QOF (S2 Fig), were similar to the results of the main analyses. Inspection of log–log survival plots by ethnic group showed no major deviations from proportionality (S3 Fig), but a formal test of Schoenfeld residuals revealed some evidence of non-proportionality for both age and ethnicity. However, given that the ethnicity hazard ratios were unaffected by the addition of interaction terms for age × time band (time bands were based on tertiles of follow-up: 0–65 days, 66–586 days, and >586 days) or ethnicity × time band, and remained similar when stratified by time band (S4 Fig), we considered the non-proportionality minor enough to justify proceeding with Cox models. Collinearity diagnostics of the main model featuring all covariates (i.e., the last model shown in Fig 1) revealed VIF ≥ 10 for BMI and TC/HDL, as well as the forced variables of age and gender. When BMI and TC/HDL were removed from the model, the resulting deflation in standard errors was minimal and did not affect the interpretation of the results; therefore, we deemed the collinearity of the model within acceptable parameters.

To assess the population impact of equalising statin prescribing across ethnic groups, we applied 2011 census ethnicity proportions [28] to the estimated UK population of people with type 2 diabetes [29] and assumed a lifetime ASCVD risk of 62% [30], giving 213,900 and 94,116 expected ASCVD events in people of South Asian and African/African Caribbean ethnicity with type 2 diabetes, respectively. Assuming that 51% of people of South Asian ethnicity with expected ASCVD events receive statins (as we have found), and a relative reduction in ASCVD risk of 37% with high-potency statin treatment [7], we would expect 40,362 ASCVD events to be prevented under current prescribing practices over the lifetimes of people of South Asian ethnicity currently affected by type 2 diabetes in the UK. For people of African/African Caribbean ethnicity (whose statin initiation rate was 41%), we would expect 14,277 prevented events over the lifetimes of people of African/ African Caribbean ethnicity currently affected by type 2 diabetes in the UK. However, applying our observed statin prescribing rate of 59% for people of

European ethnicity to the populations of South Asian and African/African Caribbean ethnicity, we would expect 46,694 and 20,545 events to be prevented. Therefore, by equalising statin initiation rates to those in the European group, an additional 6,331 and 6,268 ASCVD events, respectively, could be prevented in South Asian and African/African Caribbean people with type 2 diabetes. Using more conservative assumptions—i.e., statin adherence of 60% [31] and use of low-potency statins resulting in a 22% reduction in ASCVD events (as reported by the Heart Protection Study in diabetes, which assessed the efficacy of simvastatin 40 mg over 4.6 years of follow-up [32])—yields corresponding numbers of events prevented of 2,259 and 2,236 for people of South Asian and African/African Caribbean ethnicities, respectively. Of note, all estimated numbers of events prevented are likely to be conservative, since they are modelled on 4- to 5-year reductions in ASCVD, as reported in trials, rather than lifetime risk reductions.

## Discussion

We report markedly lower guideline-indicated statin initiation for people of African/African Caribbean ethnicity with type 2 diabetes than those of European ethnicity. A more modest reduction was seen for people of South Asian ethnicity. Even after accounting for differences in baseline cholesterol, other ASCVD risk factors, sociodemographics, and healthcare usage, statin initiation was still 24% and 9% lower, respectively, than for the European group. Moreover, time to statin initiation was longer for South Asian and African/African Caribbean groups than for people of European ethnicity.

Our finding of marked statin underprescribing for primary prevention in people with type 2 diabetes of African/African Caribbean ethnicity is commensurate with previous work from the UK [5,6] and the US [11–14]. Arguably, inequalities in prescribing may be anticipated in the US fee-for-service healthcare system, where inter-ethnic socioeconomic disparities influence health insurance coverage, and thus may place non-white ethnic groups at a disadvantage [12]. However, it is striking that we see similar levels of inequality in the UK, where healthcare is free at the point of delivery. UK studies have also reported statin prescribing to be similar [5] or higher [6] in South Asian than European people, patterns substantiated by studies of statin prescribing in non-diabetic [33] or secondary prevention populations [34,35]. Our findings for the South Asian group are not dissimilar, given that the ethnic difference was small and attenuated in the London-practice sub-group, and for the final time period of the study. Nevertheless, the distinctly lower statin initiation rate we show for the Bangladeshi population in our secondary analysis of ethnic sub-groups is discordant and should be interpreted with caution, though it may be related to the markedly younger age of diabetes onset or to the high levels of deprivation we found in this sub-group; in support of our work, several other studies report inequitable prescribing for this group [36,37]. Also in keeping with the results suggested by our secondary analyses, O'Keeffe et al. found greater levels of appropriate prescribing in London compared to most other areas of the country [38], and Finnikin et al. described greater levels of guideline-adherent prescribing after the introduction of the 2014 NICE guideline [39].

We were unable to fully account for the ethnic differences in statin initiation shown by this study despite assessing the role of sociodemographics, ASCVD risk factors, healthcare usage, comorbidity, and polypharmacy; the sole explanatory factor identified was the lower baseline cholesterol in the African/African Caribbean and South Asian groups compared to the European group. Therefore, some of the underprescription may have resulted from primary care physicians making prescribing decisions based on cholesterol value alone, rather than considering overall ASCVD risk, especially in the pre-2014 era of treating to cholesterol targets in diabetes. Nevertheless, ethnic differences were still not fully explained by adjustment for cholesterol, and we also found marked underprescription for African/African Caribbean people at

the highest tertile of TC/HDL—there was no obvious explanation for this. Further explanatory factors, such as deprivation, may have been obscured by residual confounding due to misclassification of deprivation level; the IMD is derived at area level only, thus is not a direct measure of an individual's social circumstances or wealth [23]. We could not reliably distinguish between people being offered a statin and declining, and not being offered a statin per se, and reasons for statin declinature were infrequently recorded, especially for African/African Caribbean people (4%), so it is impossible to pinpoint exact reasons for underprescribing. A portion of it may be due to patient choice, which may not be adequately informed [40]. If underprescribing is at the general practitioner's discretion, this may be due to inadequate knowledge of ethnic differences in ASCVD risk (e.g., that whilst African/African Caribbean people have lower risk of coronary disease, their risk of stroke far surpasses that of Europeans, regardless of cholesterol levels [3]); an improvement in this over time may explain the smaller ethnic differences in the last years of the study. The broader concept of structural racism, encompassing all of the factors above, is an increasingly recognised source of health inequalities [41].

Key strengths of this study include a large, nationally representative population, with individual-level electronic prescribing data—in contrast to previous UK studies, which were restricted to London practices [5] or utilised practice-level [42,43] or self-report data [6]. The innovative approach of accounting for temporal changes in national prescribing guidelines enabled us to examine prescribing differences only in the population of people with type 2 diabetes in whom guidelines indicated statin use, reducing selection bias. Unlike previous work, we adjusted for a range of individual- and practice-level covariates [5,6]. Our novel use of a longitudinal, rather than cross-sectional [5,6], design enhanced sample selection precision, improved outcome ascertainment, and allowed quantification of prescribing delays. Given that almost all UK primary care prescriptions are recorded electronically, it is likely that outcome ascertainment of statin prescriptions was near complete. However, as with all studies of electronic health records, some selection bias is likely, as more data on covariates such as BMI and smoking status are recorded for regular attenders, who in turn may have more health problems [44]. Correspondingly, our comparison of the complete case analysis sample versus those excluded due to missing data showed lower consultation rates and antihypertensive use in the latter group. Nevertheless, no differences in sociodemographics or other healthcare usage factors were found, and as this bias is likely to be equally present across ethnic groups, its impact on associations between ethnicity and prescribing is probably minimal. This study used only data from people with a recorded ethnicity of interest; we believe the complete case assumption (conditional independence between missingness and outcome) is more plausible than the missing at random assumption needed for multiple imputation, since missingness of ethnicity in primary care data is likely to be associated with ethnicity itself.

By equalising statin initiation rates between people of South Asian or African/African Caribbean ethnicity and those of European ethnicity, up to an additional 12,600 ASCVD events in people with type 2 diabetes could be prevented. Therefore, further research must urgently seek explanations for underprescribing of statins, particularly in African/African Caribbean people. If our findings are corroborated, a nationwide strategy for identifying prescribing inequities [25] and providing targeted education and prescribing interventions, followed by re-audit until equitability is achieved, is imperative and could lead to the prevention of substantial cardiovascular morbidity.

## Supporting information

**S1 Checklist. STROBE statement for the reporting of cohort studies.**
(DOCX)

**S1 Fig. Associations between ethnicity and guideline-indicated statin initiation after type 2 diabetes diagnosis: Sub-group analyses.**
(DOCX)

**S2 Fig. Associations between ethnicity and guideline-indicated statin initiation after type 2 diabetes diagnosis: Sensitivity analyses.**
(DOCX)

**S3 Fig. Log–log plot of 'survival', i.e., statin initiation, by ethnicity.**
(DOCX)

**S4 Fig. Associations between ethnicity and guideline-indicated statin initiation after type 2 diabetes diagnosis, accounting for age × time in study and ethnicity × time in study interactions.**
(DOCX)

**S1 Protocol. Independent Scientific Advisory Committee scientific protocol 19_045.**
(DOCX)

**S1 Table. Comparison of baseline characteristics of people included in the complete case analysis versus those excluded from analyses due to missing data.**
(DOCX)

**S2 Table. Baseline characteristics by TC/HDL tertile for people of African/African Caribbean ethnicity.**
(DOCX)

**S3 Table. Baseline characteristics by ethnic sub-group.**
(DOCX)

**S4 Table. Baseline characteristics of people with incident type 2 diabetes in 2006 or later with no prior statin use or ASCVD and missing ethnicity.**
(DOCX)

**S5 Table. Associations between ethnicity and guideline-indicated statin initiation after type 2 diabetes diagnosis: Missing versus European ethnicity.**
(DOCX)

**S1 Text. Ethnicity Read codes.**
(DOCX)

## Author Contributions

**Conceptualization:** Sophie V. Eastwood.

**Data curation:** Sophie V. Eastwood.

**Formal analysis:** Sophie V. Eastwood.

**Funding acquisition:** Sophie V. Eastwood.

**Investigation:** Sophie V. Eastwood.

**Methodology:** Sophie V. Eastwood.

**Project administration:** Sophie V. Eastwood.

**Supervision:** Krishnan Bhaskaran, Nishi Chaturvedi.

**Writing – original draft:** Sophie V. Eastwood.

**Writing – review & editing:** Sophie V. Eastwood, Rohini Mathur, Naveed Sattar, Liam
   Smeeth, Krishnan Bhaskaran, Nishi Chaturvedi.

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
