## [Editor Report · Decision Letter 0]

8 Feb 2021

Dear Dr Eastwood, 

Thank you for submitting your manuscript entitled "Ethnic differences in guideline-indicated statin initiation for people with type 2 diabetes: a longitudinal cohort study using the Clinical Practice Research Datalink, 2006-2019." for consideration by PLOS Medicine.

Your manuscript has now been evaluated by the PLOS Medicine editorial staff as well as by an academic editor with relevant expertise and I am writing to let you know that we would like to send your submission out for external peer review.

Please re-submit your manuscript within two working days, i.e. by February 11, 2021.

Kind regards,

Beryne Odeny

Associate Editor

PLOS Medicine

---

## [Decision Letter · Decision Letter 1]

25 Mar 2021

Dear Dr. Eastwood,

Thank you very much for submitting your manuscript "Ethnic differences in guideline-indicated statin initiation for people with type 2 diabetes: a longitudinal cohort study using the Clinical Practice Research Datalink, 2006-2019." (PMEDICINE-D-21-00613R1) for consideration at PLOS Medicine. 

Your paper was evaluated by a senior editor and discussed among all the editors here. It was also sent to independent reviewers, including a statistical reviewer. The reviews are appended at the bottom of this email and any accompanying reviewer attachments can be seen via the link below:

[LINK]

In light of these reviews, I am afraid that we will not be able to accept the manuscript for publication in the journal in its current form, but we would like to consider a revised version that addresses the reviewers' and editors' comments. Obviously we cannot make any decision about publication until we have seen the revised manuscript and your response, and we plan to seek re-review by one or more of the reviewers. 

We expect to receive your revised manuscript by Apr 15 2021 11:59PM. Please email us (plosmedicine@plos.org) if you have any questions or concerns.

We look forward to receiving your revised manuscript. 

Sincerely,

Beryne Odeny, 

PLOS Medicine

plosmedicine.org

- Please revise your title according to PLOS Medicine's style. Your title must be nondeclarative and not a question. It should begin with main concept if possible. Please place the study design ("A retrospective cohort study,") in the subtitle (i.e., after a colon). 

- Abstract summary - At this stage, we ask that you reformat your non-technical Author Summary. The Author Summary should immediately follow the Abstract in your revised manuscript. This text is subject to editorial change and should be distinct from the scientific abstract. The summary should be accessible to a wide audience that includes both scientists and non-scientists. Please see our author guidelines for more information: https://journals.plos.org/plosmedicine/s/revising-your-manuscript#loc-author-summary.

- Abstract:

1. Please structure your abstract using the PLOS Medicine headings (Background, Methods and Findings, Conclusions).

2. Please combine the Methods and Findings sections into one section, “Methods and findings”. Please ensure that all numbers presented in the abstract are present and identical to numbers presented in the main manuscript text.

3. Please include the actual amounts or percentages of relevant outcomes, not just hazard ratios or relative risks.

4. Please quantify the main results (with 95% CIs and p values).

5. Please include a summary of adverse events if these were assessed in the study.

6. In the last sentence of the Abstract Methods and Findings section, please describe the main limitation(s) of the study's methodology.

7. Abstract Conclusions: Please address the study implications without overreaching what can be concluded from the data; the phrase "In this study, we observed ..." may be useful. Please interpret the study based on the results presented in the abstract, emphasizing what is new.

- Please conclude the Introduction with a clear description of the study question or hypothesis.

- For this observational study, in the manuscript text, please indicate: (1) the analytical methods by which you planned to test your hypothesis, (2) the analyses you actually performed, and (3) when reported analyses differ from those that were planned, transparent explanations for differences that affect the reliability of the study's results. If a reported analysis was performed based on an interesting but unanticipated pattern in the data, please be clear that the analysis was data-driven.

- Did your study have a prospective protocol or analysis plan? Please state this (either way) early in the Methods section. 

- Please ensure that the study is reported according to the STROBE guideline, and include the completed STROBE checklist as Supporting Information. Please add the following statement, or similar, to the Methods: "This study is reported as per the Strengthening the Reporting of Observational Studies in Epidemiology (STROBE) guideline (S1 Checklist)." The STROBE guideline can be found here: http://www.equator-network.org/reporting-guidelines/strobe/

- Your study is observational and therefore causality cannot be inferred. Please remove language that implies causality, such as - predict, risk, effect, etc. Instead, refer to associations consistently throughout the text.

- How was race/ethnicity defined and by whom? Why was race/ethnicity considered important in this study and what it is believed to represent e.g., are SES or genetic differences being attributed to race/ethnicity?

- In statistical methods, please refer to any post-hoc corrections to correct for multiple comparisons during your statistical analyses. If these were not performed please justify the reasons. Please refer to our statistical reporting guidelines for assistance (https://journals.plos.org/plosone/s/submission-guidelines.#loc-statistical-reporting)

- Please describe how you selected your adjustment variables. 

- We note the potential for unobservable confounding in this observational study. Please consider using robust methods such as propensity score matching to address this.

- Please specify the significance level used (eg, P<0.05, two-sided) and the statistical test used to derive a p value.

- Please include line numbers in your next draft.

- Please delete the word “virtually” from the 3rd paragraph of the Results section.

Comments from the reviewers:

Reviewer #1: I confine my remarks to statistical aspects of this paper. While the general statistical approach is fine, I do have some issues to resolve before I can recommend publication.

General questions: 

Did you distinguish between people who were offered statins and refused them vs. those who never were offered them?

Has anyone looked at any ethnic based differences in the effects of statins?

Was collinearity investigated? It seems likely to be present, with this set of covariates.

p 4 - near top "leading cause of death" depends entirely on how causes are listed. Is cancer one cause or many? Is a heart attack different from a stroke? Instead, give numbers of deaths or something like that.

p. 6 - the location variable should be ONE variable with levels something like a) London b) Other England c) Other UK, The way you have it, no one can be both "London" and "other UK"

The e tables were rather oddly formatted. This is more a matter of style but I found the format a bit confusing. The cases data should be in the left most column. In table e2 what is "p for"? There are some good guidelines for tables in the American Psychological Association Style Guide (but, if these tables are OK with PLOS then I am not going to prevent publication for that reason alone).

Figure 3 was a 3D bar chart. These aren't good (see the work of William S. Cleveland). You could use a dot plot, or maybe a mosaic plot, depending on what you want to highlight.

Peter Flom

Reviewer #2: The article by Eastwood et al addresses a very relevant question regarding ethnic inequalities in health care delivery. To address this issue, the authors used the well-known real-world practice database CPRD from the UK. The final conclusion is that there is considerable underuse of statin treatment for primary cardiovascular disease prevention in people with type 2 diabetes from non-European ethnicity. The authors used sound methodology. Although the study has some limitations, including its retrospective design and other limitations associated with the design, the final message indicates that we should raise awareness among physicians and also among health care authorities. This reviewer, a non-expert in statistical analysis, did not find any methodological issues affecting the validity of the data. However, the following comments/changes should be addressed:

- The authors should include other non-UK studies both in the introduction and the discussion of the paper. The audience of this journal is an international one, and, as such, an international perspective should be adopted.

- Please, give an explanation on the reason for only including subjects without cardiovascular disease. Probably, the inclusion of subjects in secondary prevention may have yielded relevant information on ethnicity issues.

- Under Method, there is no clear explanation for using as exclusion criteria those people with missing data for TC/HDL, BMI and smoking, as explained under Results. Why did they choose these variables?

Reviewer #3: 

This is a well researched and well written paper with some major and minor issues. I think the handling of missing values and the inclusion of three NICE time periods in one study are the major issues that need to be better justified.

1. 1st bullet point in the "What this study adds" section and Conclusions in the Abstract has 25%. It is not clear where this number came from.

2. 3rd bullet point in the "What this study adds" section reads more like a conjecture, not what this study adds to the body of science. Please revise.

3. Page 4, Introduction - "governed by the NICE guidelines". This is ambiguous. Is it recommended for all patients with T2D? In what strength? For example, American Diabetes Association recommends moderate intensity statins to all persons with diabetes over age 40 years. 

4. Page 5, first paragraph. How were the guidelines different between versions in terms of statin recommendations. Table 1 has this information. A brief summary of the changes in statin recommendations in 

the introduction will be very helpful. I think this is critically important for your readers to understand the baseline for the time to statin initiation you used in the study. I am also curious about whether combining the three periods in one analysis is valid. Because the recommendations changed over time, for example, all people aged 40 y or older or ASCVD 10-year risk >= 10% in NICE 2008, how was your inclusion criteria differed from NICE 2006 period or NICE 2014 period? There should be more justification about combining the three periods. Sensitivity analyses on this issue would be very helpful.

5. Page 6. Before Covariates, I think you need a section on how your events were defined and how you computed the time to event. It seems that your baseline was changing over time depending on the NICE guidelines. Your time to event calculation needs to be well justified.

6. Page 6, "For HbA1c and BMI, we also used ..." This sentence is redundant.

7. Page 6, last paragraph. How were people with harmful side effects of statins handled? What was the percentage of people with non-statin cholesterol-lowering medications? No excluding these patients from the analysis may have confounded the study results. 

8. Page 7, top of the page. What was the missing rates for the ethnicity variable? If the rate is high, the complete case analysis may be biased. Also, as the authors described it, if missing values were not MAR, complete case analysis is even more troubling. Can you present analysis based on all records by assigning missing values in a separate category whenever possible? For example, missing ethnicity in the "unknown" category.

9. Page 7, middle of page. "Multi-level Cox ..." There is not justification for using multi-level model or any discussion about how the model was used. What was the cluster/group variable? Was it a random intercept model or something else?

10. Page 11, predicted reduction in ASCVD risk is very helpful. Why not use this information for a bullet point in the "What this study adds"?

11. Page 12, bottom of the second paragraph. I do not agree with what was asserted here. Comparison of complete case sample with the entire sample in terms of covariates does not guarantee that the complete case analysis is not biased. 

12. Page 12. Can you move the key strengths paragraph somewhere to the bottom of the discussion?

[LINK]

---

## [Decision Letter · Decision Letter 2]

14 May 2021

Dear Dr. Eastwood,

Thank you very much for re-submitting your manuscript "Ethnic differences in guideline-indicated statin initiation for people with type 2 diabetes: a retrospective cohort study using the Clinical Practice Research Datalink, 2006-2019." (PMEDICINE-D-21-00613R2) for review by PLOS Medicine.

I have discussed the paper with my colleagues and the academic editor and it was also seen again by two reviewers. I am pleased to say that provided the remaining editorial and production issues are dealt with we are planning to accept the paper for publication in the journal.

[LINK]

We look forward to receiving the revised manuscript by May 21 2021 11:59PM.   

Sincerely,

Beryne Odeny, 

Associate Editor 

PLOS Medicine

plosmedicine.org

Requests from Editors:

1. Please revise your title to include the country/setting

2. Please trim your author summary bullet points

3. Please add the following statement, or similar, to the Methods: "This study is reported as per the Strengthening the Reporting of Observational Studies in Epidemiology (STROBE) guideline (S1 Checklist)."

4. Please provide the meaning of abbreviations in tables (Table S3, S4, S5 e.g., TC, HDL, BMI, CKD, COPD, NICE, HbA1c

5. Please provide 95% CIs and p values for all estimates in the main text and tables.

6. Thank you for providing your STROBE checklist. Please replace the page numbers with paragraph numbers per section (e.g. "Methods, paragraph 1"), since the page numbers of the final published paper may be different from the page numbers in the current manuscript.

7. Please use the "Vancouver" style for reference formatting and see our website for other reference guidelines https://journals.plos.org/plosmedicine/s/submission-guidelines#loc-references . Please ensure that weblinks are current and accessible

Comments from Reviewers:

Reviewer #1: The authors have addressed my concerns and I now recommend publication.

Peter Flom

Reviewer #2: The authors have properly addressed the questions raised by this reviewer

Reviewer #3: This revised version satisfactorily addressed all of my comments. I do not have any more concerns.

[LINK]

---

## [Editor Report · Decision Letter 3]

20 May 2021

Dear Dr. Eastwood,

Thank you very much for re-submitting your manuscript "Ethnic differences in guideline-indicated statin initiation for people with type 2 diabetes in UK primary care: a retrospective cohort study using the Clinical Practice Research Datalink, 2006-2019." (PMEDICINE-D-21-00613R3) for review by PLOS Medicine.

I have discussed the paper with my colleagues and the academic editor and it was also seen again by three reviewers. I am pleased to say that provided the remaining editorial and production issues are dealt with we are planning to accept the paper for publication in the journal.

[LINK]

We look forward to receiving the revised manuscript by May 27 2021 11:59PM.   

Sincerely,

Beryne Odeny, 

Associate Editor 

PLOS Medicine

plosmedicine.org

Requests from Editors:

Thank you for your revisions. Please make this final set of revisions before acceptance:

1. Please trim the title and modify it to, “Ethnic differences in guideline-indicated statin initiation for people with type 2 diabetes in UK primary care, 2006-2019: A cohort study”

2. Please provide the meaning of the abbreviation "HR" in Figure S4

3. Please correct reference #31 and remove excess text after the citation

4. Please edit #40 and remove the paragraph on “www.icmje.org/coi_disclosure.pdf and declare:”

Comments from Reviewers:

[LINK]

---

## [Editor Report · Decision Letter 4]

25 May 2021

Dear Dr Eastwood, 

On behalf of my colleagues and the Academic Editor, Dídac Mauricio, I am pleased to inform you that we have agreed to publish your manuscript "Ethnic differences in guideline-indicated statin initiation for people with type 2 diabetes in UK primary care, 2006-2019: A cohort study" (PMEDICINE-D-21-00613R4) in PLOS Medicine.

PRESS

Sincerely, 

Beryne Odeny 

Associate Editor 

PLOS Medicine